# Genetic Transformation of the Model Quorum Sensing Bacterium *Vibrio campbellii* by Electroporation

**DOI:** 10.3390/genes16060626

**Published:** 2025-05-24

**Authors:** Tanya Tschirhart, Zheng Wang, Dagmar H. Leary, Gary J. Vora

**Affiliations:** US Naval Research Laboratory, Center for Bio/Molecular Science and Engineering, Washington, DC 20375, USA; tanya.tschirhart.civ@us.navy.mil (T.T.);

**Keywords:** aquaculture, bioluminescence, extracellular nuclease, *Vibrio harveyi*

## Abstract

Background: The marine bacterium *Vibrio campbellii* has been a model system for the study of bacterial quorum sensing and is increasingly recognized as a formidable aquatic animal pathogen. While genetically tractable, the study of this species in basic and applied research still relies upon laborious and time-consuming conjugation methods for plasmid DNA transformation. Methods: In this study, we developed an electroporation protocol using the most studied strain of this species, *V. campbellii* ATCC BAA-1116. An electroporation efficiency of up to 3 × 10^4^ CFU/μg DNA was demonstrated using derived parameters (10 kV/cm, 400 Ω, 25 μF), which took cell growth phase at harvest, plasmid DNA amount, and recovery conditions into account. The electroporation protocol was tested using several different plasmids and with additional strains of *V. campbellii* and sister species *V. harveyi*. Results: Interestingly, of the eight other *V. campbellii* strains tested, only three others, which also happened to be the three most recent environmental isolates with the fewest number of laboratory passages, were amenable to electroporation-mediated transformation. Conclusions: This electroporation protocol expands the tool set for studying *V. campbellii* and provides interesting insights into DNA transformation and uptake in this and related bacterial species.

## 1. Introduction

The marine bacteria *V. campbellii* and *V. harveyi* are members of the Harveyi clade [1] and have found a unique niche within the research community as important model systems. Within basic research contexts, the most-studied characteristic of *V. campbellii* and *V. harveyi* is quorum-sensing and its resulting phenotypes: bioluminescence, biofilm formation, and virulence [2,3,4]. This research has led to insights regarding specialized natural functions [5,6,7], bacterial communication in natural communities [8], and forays to interfere with quorum sensing mechanisms to regulate pathogenicity [9,10]. Within applied research contexts, *V. campbellii* and *V. harveyi* are two of the most economically important species in this clade as known pathogens of many commercially farmed marine animals [11]. For example, pathogenic members of these species cause vibriosis and are being increasingly recognized as agents causing mass mortality events in shrimp aquaculture [12].

*V. campbellii* strain ATCC BAA-1116 (also known as BB120) [13] has been the model strain of this species. BAA-1116 has a fully sequenced genome, well-defined quorum sensing system, and is genetically tractable. While an impressive amount of knowledge on gene regulation, signal transduction, and quorum sensing-induced phenotypes has been generated via the ability to transform plasmid DNA into *V. campbellii* BAA-1116, all such manipulations have required the use of laborious and time-consuming conjugation methods [5,14]. Electroporation is a significantly faster method for DNA transformation than conjugation and conveys other advantages, such as the ability to introduce linear DNA into cells for recombineering and other purposes. An efficient electroporation-mediated transformation protocol would accelerate and facilitate efforts to better understand the cellular impacts of quorum sensing, pathogenicity mechanisms, extracellular products, and genomic stability on this model system. Additionally, since electroporation requires less overall passaging of cells, there should be fewer genetic differences in the transformed cells and less genetic deterioration from the original strain. This is especially important as vibrios are known to be genetically promiscuous.

Marine bacteria present a specific set of challenges when it comes to the development of an electroporation method for DNA transformation. First, as inhabitants of the marine environment, these bacteria not only prefer but often require high salinity for growth and osmotic balance. However, any presence of salt is detrimental to electroporation as high conductivity of the buffer and cells can result in arcing. Therefore, one challenge has been to determine a buffer that would be osmotically balanced while also not interfering with electroporation. The use of high millimolar concentrations of sucrose or sorbitol has been demonstrated to solve this problem in other *Vibrio* species or marine bacteria with similar characteristics [15,16,17,18,19,20]. Another challenge is the fact that some vibrios are known to have extracellular or periplasmic DNAses, which could prematurely degrade plasmid DNA that is introduced through transformation [21,22]. To counter this issue, methods have been developed to use buffers lacking Mg^2+^ ions to decrease or prevent the function of these enzymes (which require Mg^2+^) [15,16] or have introduced an osmotic shock prior to electroporation to purge the periplasm of DNases [17].

In this study, we developed a method that reliably enables plasmid DNA transformation through electroporation in multiple *V. campbellii* strains. We present optimized parameters for cell preparation, electroporation conditions, and cell recovery, and demonstrate the electroporation and function of several plasmids with various origins of replication and selectable resistance markers.

## 2. Materials and Methods

### 2.1. Strains and Plasmids Used in This Study

The *V. campbellii* strains used in this study were: ATCC BAA-1116, HY01, DS40M4, PEL22A, E1, ATCC 25940 (type strain), CAIM 1500, BoB-53, and BoB-90. *V. harveyi* strains used in this study were: ATCC 14126 (type strain), 392 MAV, and ATCC 35084 (Appendix A). The plasmids used in this study and their relevant characteristics are listed in Table 1.

### 2.2. Media and Culture Conditions

Cells were cultured in LM media (25 g Luria Broth powder, 10 g NaCl, in 1 L) (Sigma- Aldrich, St. Louis, MO, USA). Solid media utilized 1.5% agar (Sigma-Aldrich). LM agar plates were used for colony-forming unit (CFU) counting and selection. For recovery after electroporation, typically, SOC media (Thermo Fisher Scientific, Waltham, MA, USA) with an additional 0.51 M NaCl was used. Where indicated, sucrose (EMD) at 400 mM or 680 mM was added. BHI media was prepared from powder as indicated by the manufacturer (Teknova, Hollister, CA, USA). LBv2 media consists of Luria Broth media with additional salts (204 mM NaCl, 4.2 mM KCl, 23.14 mM MgCl_2_) (Sigma-Aldrich). The antibiotics used were kanamycin sulfate and chloramphenicol (Sigma-Aldrich), as well as tetracycline hydrochloride. Typical culture conditions were 30 °C with 225 rpm shaking in liquid media or 30 °C without shaking on solid media. Where indicated, 1 mM IPTG (Sigma-Aldrich) was used for induction. Diaminopimelic acid (DAP) was from Thermo Scientific Chemicals (Carlsbad, CA, USA).

### 2.3. Electrocompetent Cell Preparation and Transformation by Electroporation

A 2 mL culture of cells in LM media was grown overnight at 30 °C, 225 rpm. The overnight culture was inoculated at 1% into a 250 mL flask with 35 mL LM. The cells were cultured at 30 °C, 225 rpm until the OD_600_ was ~0.3–0.5, which for most strains, took no longer than 2 h. Cells were chilled on ice in pre-chilled 50 mL tubes for 30 min. In a pre-chilled (3–4 °C) centrifuge, the cells were pelleted at 4000× *g* for 20 min. The supernatant was carefully decanted, and the pellet was gently resuspended in 30–50 mL of sterile-filtered, ice-cold 400 mM sucrose (or other, as indicated) in Milli-Q water. The cells were then re-pelleted at 4500× *g* for 15 min. The supernatant was removed by pipette, and the pellet was resuspended in half the volume of ice-cold 400 mM sucrose (or other, as indicated). The wash step was repeated twice, for a total of three washes. Afterwards, cells were resuspended in the residual buffer. The OD_600_ was measured and, if necessary, adjusted to ~40. Aliquots of 50 µL were used right away or frozen at −80 °C for future use.

For electroporation, DNA was added to the cells in a total of 3 µL of nuclease-free water, transferred to a pre-chilled 1 mm electroporation cuvette, and electroporated at the indicated voltage, capacitance, and resistance settings. For recovery, 500–1000 µL of the indicated recovery media (typically SOC + 0.51 M NaCl) was immediately added to the cells, which were recovered at 30 °C for the indicated time (typically 1.5 h). Between 50 and 200 µL of recovered cells were plated on pre-warmed LM agar plates with the appropriate antibiotics. Plates were placed at 30 °C overnight. Transformed colonies were visible after 16 to 48 h.

### 2.4. Transformation by Conjugation

Conjugation was performed by mating *V. campbellii* with the *E. coli* WM3064 DAP auxotroph. First, *E. coli* WM3064 with the desired plasmid and *V. campbellii* were plated on LB + DAP + antibiotic at 37 °C or on LM at 30 °C overnight, respectively. A single colony was selected from each plate and grown in liquid LB + antibiotic + DAP at 37 °C and 250 rpm or in LM at 30 °C and 250 rpm for *E. coli* WM3064 or *V. campbellii*, respectively. The following day, 3 mL of WM3064 was inoculated at 2% in LB with antibiotics and DAP and grown for about 3 h. *V. campbellii* was inoculated simultaneously into 3 mL of LM and grown at 30 °C for 4 h. Following this, 300 µL of the WM3064 and *V. campbellii* were each spun down at 8000 rpm for 3 min in a tabletop centrifuge and washed in 500 µL of LM media. The pellets were resuspended in 200 µL LM. For the conjugation, 100 µL of each resuspended culture was mixed in a tube, and both the mixed culture and separate cultures were spotted onto a LM + DAP plate and allowed to dry. This plate was incubated at 30 °C overnight, after which the bacterial lawn was collected and resuspended in 1 mL LM. Several serial dilutions were done in LM and plated onto LM selection plates without DAP. Colonies were observed and tested the next day.

### 2.5. Determination of Natural Antibiotic Resistance Levels and Plasmid Function

To determine which plasmids could be used for electroporation, we first tested natural antibiotic resistance in the various strains under study. LM agar plates with various concentrations of the antibiotics were prepared, and 3 µL of a 1:10^6^ dilution of an overnight culture was spotted on the plate. Plates were incubated at 30 °C overnight and checked for growth. An additional day of growth was then allowed, and plates were checked for colonies again. *V. campbellii* BAA-1116 growth was inhibited at 150 µg/mL kanamycin, though sometimes spontaneous mutants arose, and small colonies could be seen on the second overnight growth. BAA-1116 was also inhibited by 2 µg/mL of chloramphenicol and 2 µg/mL tetracycline. We used 150 µg/mL kanamycin, 6 µg/mL chloramphenicol, and 2 µg/mL tetracycline for the remainder of this study. The other strains were also partially analyzed for resistance, and the results are reported below. For plasmids pJV315, pJV298, and pJV021, plasmid presence was confirmed by overnight induction of GFP with 1 mM IPTG from mid-log phase. Cell pellets from induced cultures were visibly green. Plasmids with the oriT transfer origins were first conjugated into *V. campbellii* (through WM3064, as indicated above) to verify their ability to replicate and to confirm antibiotic resistance levels.

### 2.6. Extracellular Proteomics

*V. campbellii* BAA-1116, *V. campbellii* HY01, and *V. harveyi* 392 MAV were inoculated from glycerol stocks into 3 mL of LM medium and cultured overnight at 250 rpm and 30 °C. Each culture (1 mL) was then used to re-inoculate 50 mL of LM medium in a 250 mL flask. Cells were grown at 250 rpm until the OD_600_ was between 0.43 and 0.55. Each flask’s culture was split into three 50 mL Falcon tubes, with ~16 mL each. For each strain, one aliquot was kept as is, and ~360 ng of genomic DNA was added to the other two (this was adjusted for the OD_600_ of the cells to provide equimolar DNA amounts per cell). One of the aliquots with added DNA was put back in 30 °C for 3 min, and the other for 2 h. After the elapsed time, the cells were pelleted (8000× *g* for 5 min at room temperature) and the supernatant was saved for analysis. The aliquot without DNA added was pelleted without delay or incubation at 30 °C.

Extracellular protein-containing supernatants (12.5 mL) were carefully removed from each aliquot, filtered through MWCO 10 kDa membrane filters, and washed with 50 mM ammonium bicarbonate. Once supernatant volumes were reduced to 200–400 µL, they were transferred onto smaller pore filters (MWCO 3 kDa) and further washed with 50 mM ammonium bicarbonate. Once reduced to 100 µL volumes, 10% n-propanol in 50 mM ammonium bicarbonate was added for in-solution trypsin digestion. The concentrated extracellular protein samples were digested in a HUB-440 Baro-cycler (Pressure Biosciences, Inc., Canton, MA, USA) for 60 cycles (20 s ON, 10 s OFF, 45 kpsi at RT). Digests were frozen and dried under vacuum prior to analysis via LC-MS/MS using a U3000 Thermo LC coupled to an Orbitrap Fusion Lumos. Resulting spectra were extracted and searched by Mascot against the appropriate FASTA database for each *Vibrio* strain. Proteins identified by Mascot Server (v.2.8.2; Matrix Science, London, UK) were exported into comma-separated files and inspected for the identification of DNA-degrading enzymes. The term “nuclease” was used to search the names of all identified proteins. The FASTA protein databases corresponding to each *Vibrio* strain (containing all predicted proteins translated from genome sequences *in silico*) were also searched for the term nuclease to determine the total number of encoded predicted/possible nucleases by the genome of each strain. Additionally, direct sequence comparison to DNA-specific endonuclease I (Dns) of *V. natriegens* 14048 (ALR14600) via the BLAST algorithm was carried out *in silico*. Deletion of ALR14600 was demonstrated to have increased transformation efficiency [20].

## 3. Results

### 3.1. Electroporation Parameter Testing

Electroporation parameters were first studied with plasmid pVSV105 [23] in *V. campbellii* ATCC BAA-1116. First, we varied the electric field strength between 5–15 kV/cm, with the resistance set at 200 Ω and capacitance at 25 µF. Results from duplicate experiments demonstrated that 10 kV/cm provided the highest transformation efficiency (Figure 1a). This field strength was then chosen, and varying resistances were tested (200, 400, 600, and 800 Ω), with 150 ng DNA and 25 µF. Results from triplicate experiments demonstrated that 400 Ω provided the highest transformation efficiency, though 200 and 600 Ω resistances were not significantly different (Figure 1b). Under these conditions, we observed ~64% cell viability after electroporation based on CFU counts.

We then tested varying plasmid DNA amounts (between 10 and 500 ng), all in 3 µL of nuclease-free water. We performed these experiments several times and with different batches of plasmid DNA and electrocompetent cells. Interestingly, lower DNA concentrations resulted on average in higher transformation efficiencies (Figure 1c). In all cases, DNA was purified from DH5α or Top10 *E. coli*. Electroporation of DNA purified from previously transformed (through conjugation or electroporation) *V. campbellii* ATCC BAA-1116 cells did not show a significant difference in efficiency, indicating that any differences in methylation patterns do not play a significant role in this case.

Recovery time after electroporation can have a significant effect on transformation efficiency, especially with comparatively slower-growing bacteria like *V. campbellii*. We investigated recovery at 45 min, 90 min, and 180 min and noted higher efficiency with longer recovery time, as well as the previously observed trend of higher efficiency with lower DNA input (Figure 1d). Although there did seem to be more transformed colonies using 180 min of recovery time, it also increased the experimental time significantly; therefore, 90 min was selected as a reasonable recovery time. However, for other strains, 180 min was required to observe a significant number of colonies (see below).

### 3.2. Effects of Cell Preparation and Media on Transformation Efficiency

Once the electroporation conditions were set, we optimized several aspects of cell preparation before or after electroporation: cell concentration (OD_600_), growth phase, wash and electroporation buffers, and recovery media. First, we harvested cells at mid-log phase, washed with 400 mM sucrose, and prepared them at either 1× or 2× concentration (OD_600_ 21 or 42). While previous reports have shown successful electroporation of *Vibrio* when they are highly concentrated [26], as a practical consideration, using fewer cells would result in a greater number of aliquots of electrocompetent cells. For most DNA concentrations utilized, the higher-OD_600_ cell concentration demonstrated higher transformation efficiencies (Figure 2a). Interestingly, the efficiency when using 25 ng DNA was similar between the two conditions tested. Therefore, we concluded that for most applications, cells should be concentrated on an OD_600_ of ~40 for best results.

Next, we studied cells that were collected after growth to an OD_600_ of 0.35 or 1. In previous studies, *V. alginolyticus* and *V. vulnificus* grown to late exponential to early stationary phase ODs were more efficiently transformed via electroporation [17,19]. The cells were washed and prepared as before, both finally re-suspended to an OD_600_ of ~40, and electroporated using the optimized conditions with different amounts of DNA. Again, a trend where lower DNA amounts result in higher efficiencies for both cell preparations was observed (Figure 2b). There was also a significant overall decrease in efficiency with cells that were collected at a later stage of growth. We concluded that cells should be collected at mid-log phase, as is typical in many other bacterial transformation protocols.

We tested one additional wash and electroporation buffer, 680 mM sucrose. This electroporation buffer was recently used to prepare electrocompetent *V. natriegens* [20], which is closely related to *V. campbellii* and also a member of the Harveyi clade. Here, we found that the *V. campbellii* cells were much harder to resuspend during the wash steps than in 400 mM sucrose (this is true for preparations of *V. natriegens* as well). However, in this case, transformation efficiency suffered as well, even when varying electric fields and resistances. The maximum transformation efficiency with 680 mM sucrose buffer was approximately 25% of the efficiency reached in 400 mM sucrose buffer (Figure 2c).

In addition to the electroporation medium, the recovery medium composition can also play a role in transformation efficiency. In most of our studies, we used SOC with an additional 0.51 M NaCl. However, successful protocols in use for *V. natriegens* use BHI + v2 salts [20]. Therefore, we tested BHI + v2 salts, LB + v2 salts, and SOC + 0.51 M NaCl. The choice of recovery medium did not have a significant effect on electroporation efficiency under these conditions (Figure 2d). Although with other strains there was a more positive effect from adding the v2 salts (see below), taking all facets into account, we concluded that the use of mid-log-phase cells, washed in 400 mM sucrose, concentrated to an OD_600_ of approximately 40, and recovered in SOC + NaCl, provided the best results. For ease of implementation, a schematic of the optimized protocol (from initial cell culture to electroporation with plasmid DNA and recovery) is summarized in Figure 3.

### 3.3. Electroporation of Different Plasmids

Next, we explored the electroporation of different plasmids with different origins of replication, antibiotic resistance markers, and sizes (Table 1). We successfully electroporated plasmids with the pES2113 and p15A origins of replication into *V. campbellii* ATCC BAA-1116 (i.e., pJV315, pJV298, pJV021, pVSV102, and pVSV05). We were unable to transform the pSEVA237R or pSEVA351 vectors [25] into any *V. campbellii* or *V. harveyi* strains. Whenever possible, plasmids with chloramphenicol resistance were used since most strains were sensitive to the antibiotic (with a few exceptions). Kanamycin or tetracycline resistance gene-containing plasmids were used in other cases when the strains were resistant to chloramphenicol (up to 20 µg/mL).

### 3.4. Electroporation of Other V. campbellii and V. harveyi Strains

In addition to *V. campbellii* BAA-1116, we prepared additional strains of *V. campbellii* and its sister species *V. harveyi* using the optimized protocol and electroporated using a subset of the originally tested parameters. We used the plasmids pJV298 (or pJV021 for Cmp-resistant strains) [24] and pVSV105 (or pVSV102 for Cmp-resistant strains), as they have shown the most consistent and highest transformation efficiencies. No transformants were obtained for *V. campbellii* HY01, DS40M4, E1, and CAIM 1500 and *V. harveyi* strains 392 MAV, ATCC 35084, and ATCC 14126 under any of the conditions tested. However, we were able to obtain and confirm *V. campbellii* PEL22A transformants using 25 ng of pJV298. The protocol adjustment for PEL22A required a recovery of 180 min, 0.75 kV, and 200 Ω resistance, and gave a maximum efficiency of 4.53 × 10^3^ CFU/µg DNA for the pJV298 plasmid, but no transformation of the pVSV105 plasmid. We were also able to confirm transformants of *V. campbellii* BoB-90 with pVSV105, but not pJV298, plasmid using a recovery time of 180 min, 1 kV, and 400 Ω in the LB + v2 salts recovery medium. The maximum efficiency was 3.9 × 10^3^ CFU/µg DNA. *V. campbellii* BoB-53 was also successfully electroporated with pJV021 but not the pVSV102 plasmid with a recovery time of 180 min, 1 kV, and 400 Ω in SOC + NaCl recovery medium. The maximum efficiency was 5 × 10^3^ CFU/µg DNA.

### 3.5. Possible Extracellular Nucleases

We investigated whether the presence of known or DNA-inducible extracellular nucleases could help explain the inability to electroporate certain strains using *V. campbellii* HY01, *V. campbellii* BAA-1116, and *V. harveyi* 392 MAV. Of these three, BAA-1116 is the only strain that was successfully transformed using our protocol. Cell cultures for all three strains were collected at mid-log phase growth and divided into three treatment groups ((1) no exogenous DNA addition (0 min); (2) ~360 ng of genomic DNA added for 3 min; (3) ~360 ng of genomic DNA added for 2 h), from which cell-free supernatants were processed for LC-MS-based proteomics. We identified a comparable number of unique peptides belonging to “nucleases” in each of the three treatment groups tested (Figure 4a, Appendix A) with two notable trends: *V. harveyi* 392 MAV yielded more unique peptides from nucleases post-DNA pulse, whereas *V. campbellii* BAA-1116 demonstrated a decrease in unique peptides from nucleases over time. Within this data, we also searched for the presence of DNA-specific endonuclease I (Dns) homologs. In the sister species *V. natriegens*, strains with a *dns* gene knockout display increased transformation efficiencies and are routinely used by researchers in cloning and synthetic biology applications [20,27]. BLAST (version 2.16.0) searches confirmed that homologs of the *V. natriegens* Dns protein (ALR14600) are encoded by the *V. campbellii* HY01, *V. campbellii* BAA-1116, and *V. harveyi* 392 MAV genomes (Appendix A), and these homologs were indeed detected in the supernatants tested. Exponentially modified protein abundance index (emPAI) quantitation assessments identified the Dns homologs in all three strains to be among the top three nucleases detected in each treatment group. Interestingly, the abundance of the Dns homologs increased in the two strains that we could not successfully transform via electroporation (*V. campbellii* HY01 and *V. harveyi* 392 MAV) after the addition of exogenous DNA (Figure 4b). Conversely, in one of the strains, we were able to successfully electroporate, *V. campbellii* BAA-1116; the Dns homolog was undetectable 2 h after the addition of exogenous DNA. One interpretation of these results is that exogenous DNA induces the expression of the Dns homolog in *V. campbellii* HY01 and *V. harveyi* 392 MAV, thus degrading the extracellular DNA that was to be transformed, whereas the non-inducible nature of this enzyme by exogenous DNA in *V. campbellii* BAA-1116 enables successful electroporation-based DNA transformation.

## 4. Discussion

As notable pathogens of many commercially farmed marine animals, *V. campbellii* and *V. harveyi* are known to cause substantial economic losses for the aquaculture industry annually. Given their impact, it is surprising that the study of their pathogenicity mechanisms and the development of control measures have been slowed in part by the absence of a basic tool for genetic manipulation, electroporation. To date, Delavat et. al. have provided the only report of successful transformation of plasmid DNA through electroporation in any *V. harveyi* or *V. campbellii* strain [26]. The developed one-day-long electroporation protocol was sufficient to transform two *V. harveyi* strains (ORM4 and LMG 7890), two *Pseuodoalteromonas* strains (MV21 and 3 J6), and also *V. campbellii* ATCC BAA-1116. Specifically in the case of *V. campbellii* ATCC BAA-1116, the protocol was able to transform a 4.8 kb plasmid with an electroporation efficiency reaching 2.7 × 10^2^/µg DNA [26].

It is interesting to note that in all of our experiments, we observed higher transformation efficiencies when lower amounts of plasmid DNA were used. This is the opposite of what is typically observed for many other bacterial species, including several other *Vibrio* [15,17]. Interestingly, the commercially available *V. natriegens*-based Vmax™ cells (Telesis Bio, Inc, San Diego, CA, USA) also indicate the preferred use of 5 ng of DNA or less for electroporation, suggesting that this Harveyi clade sister species may also have higher efficiencies of electroporation with lower DNA amounts. Additionally, *V. vulnificus* shows a similar trend when electroporated with the shuttle vector PVv3 [19]. It is possible that higher levels of DNA trigger the release of nucleases, and our work begins the exploration of this possibility. Overall, this could be an interesting characteristic of these bacteria that can better inform future method development for transformation and genetic manipulation.

We have demonstrated that this protocol can be adapted for the transformation of other *V. campbellii* strains. While the specific parameters will likely need to be optimized for each, this effort provides a good starting point from which to elaborate. The fact that some of the strains tested were able to maintain electroporated plasmids with only certain origins of replication highlights the importance of testing plasmids with different origins in new strains. Since we did not test the compatibility of every origin of replication in each strain, a lack of transformation may actually reflect origin incompatibility rather than electroporation efficiency. Thus, it is possible that in the strains where electroporation was unsuccessful, an origin of replication, which we did not test, was required for successful maintenance. Antibiotic counter selection plays an important role in deciding which plasmids to use for testing, and chloramphenicol was the choice for this study. Many vibrios, including most we utilized here, are ampicillin- and kanamycin-resistant. Tetracycline can be chelated by higher salt concentrations in growth media (which vibrios require) and therefore is less bioavailable and effective for this application.

As mentioned previously, many marine bacteria are known to produce DNases that are present in the periplasm or secreted extracellularly [28], and these can result in notably lower transformation efficiency. One salient example is that of the Dns and Xds extracellular nucleases from *V. cholerae* [22], and it has been demonstrated that removing the Dns nuclease results in significantly higher transformation efficiencies [29,30]. Based on our LC-MS-based characterization of the extracellular proteome, this may well be the case for *V. campbellii* as well and would suggest Dns deletion as a general strategy for the development of DNA electroporation protocols. A related option could be the depletion of Dns during competent cell preparation by subjecting the cells to osmotic shock to remove DNases from the periplasm [17]. A third potential solution that also does not require genetic manipulation is to use highly concentrated samples of DNA (>550 ng) to overwhelm these nucleases, or the use of carrier DNA or other methods to mask the plasmid DNA of interest.

The facile and efficient electroporation of single plasmids, plasmid libraries, and linear DNA into bacteria is an important tool to facilitate genome screening and engineering workflows. The electroporation of pooled plasmid libraries containing CRISPR-Cas9 guide RNA, for example, could be used to screen genes that impact a specific phenotype of interest in *V. campbellii*. A logical follow-on to the development of an electroporation method is the electroporation of linear DNA for genome engineering purposes. Recombineering methods rely on the ability of linear DNA to enter the cell and integrate into the chromosome at specific locations. This linear DNA can be introduced via electroporation but not via conjugation methods. Recombineering methods for *V. campbellii* (perhaps based on a recently developed method for *V. natriegens* [31]), facilitated by the method described herein, may result in an even more useful tool for genetic manipulation and the study of this species.

## 5. Conclusions

We have developed a simple protocol for the electroporation of the model quorum-sensing bacterium *V. campbellii* ATCC BAA-1116 with a transformation efficiency of up to 3 × 10^4^ CFU/µg DNA. It is anticipated that this method will save experimental time, abrogate the need to subculture cells for conjugation, and provide a valuable starting point for the development of protocols for recalcitrant or newly discovered strains. It was both interesting and encouraging that the developed protocol enabled the manipulation of our three most recently collected environmental isolates, suggesting that this protocol can facilitate the study of currently circulating and non-laboratory-adapted strains.

## Figures and Tables

**Figure 1 genes-16-00626-f001:**
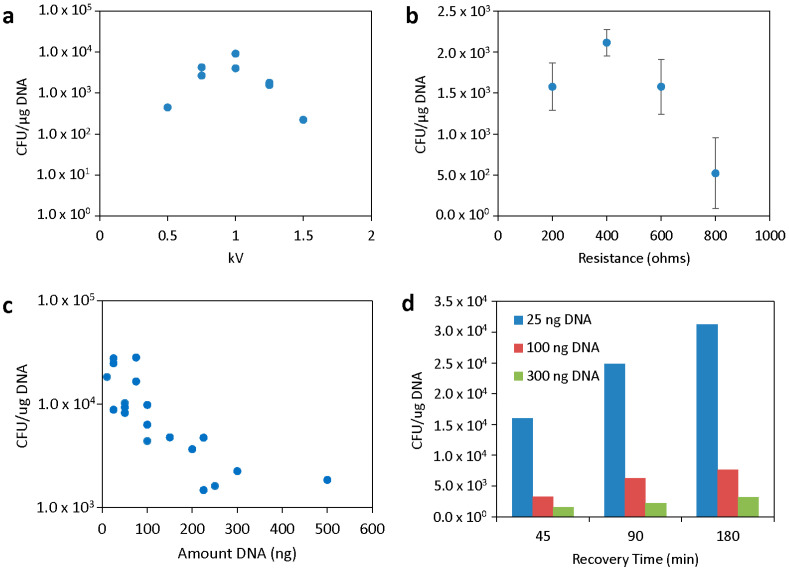
Optimization of electroporation conditions. (**a**) Optimization of the electroporation voltage. Resistance was constant at 200 ohms, capacitance at 25 µF, and 150 ng of plasmid DNA was used. (**b**) Optimization of electroporation resistance. Voltage was constant at 1.25 kV, capacitance at 25 µF, and 150 ng of plasmid DNA was used. (**c**) Transformation efficiency after different amounts of plasmid DNA were electroporated with 1 kV, 400 ohms, and 25 µF. (**d**) Transformation efficiency after different amounts of plasmid DNA were electroporated with the same conditions as in (**c**) and plated after different recovery times. The plasmid pVSV105 was used throughout.

**Figure 2 genes-16-00626-f002:**
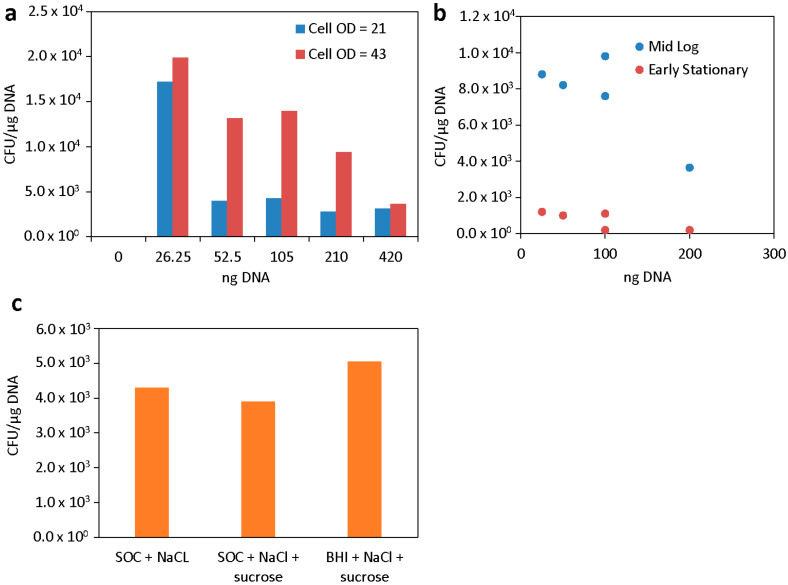
Optimization of cell preparation conditions. (**a**) Effect of apparent cell OD_600_ on electroporation efficiency. (**b**) Effect of cell growth phase at harvest on electroporation efficiency. (**c**) Effect of electroporation/wash medium on electroporation efficiency. (**d**) Effect of recovery media on electroporation efficiency of 100 ng plasmid pJV298. For all experiments in this figure, 1 kV, 25 µF, 400 ohm, and 90 min recovery time were used. SOC + NaCl was used unless indicated otherwise.

**Figure 3 genes-16-00626-f003:**
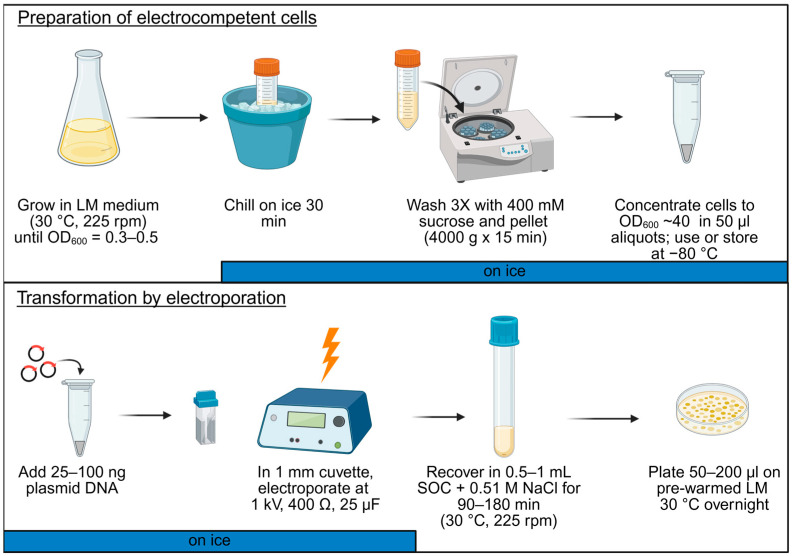
Summary schematic of electrocompetent cell preparation and transformation by electroporation for *V. campbellii* BAA-1116. This graphic was created in BioRender.

**Figure 4 genes-16-00626-f004:**
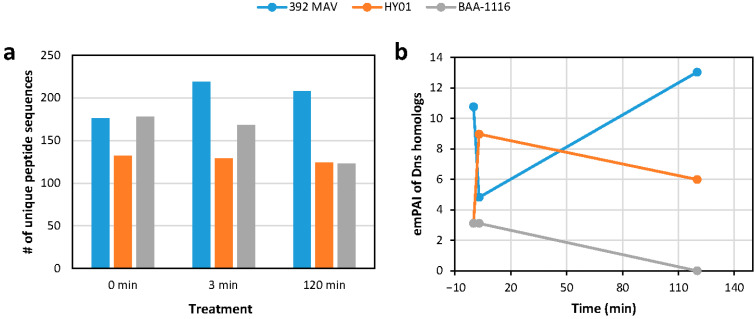
Extracellular nuclease identification via LC-MS. (**a**) Number of unique peptides belonging to nucleases detected in each strain and treatment. (**b**) EmPAI relative abundance assessments of the Dns nuclease homologs over the three treatments.

**Table 1 genes-16-00626-t001:** Plasmids used in this study.

Plasmid	Features	Size (bp)	Source
pVSV105	pES213 ori, oriT, RK6, Cm	5780	Ref. [23]
pVSV102	Constitutive GFP, pES213 ori, oriT, RK6, Kan	6440	Ref. [23]
pJV021	IPTG-inducible GFP, p15A ori, oriT, Kan	5017	Ref. [24]
pJV298	IPTG-inducible GFP, p15A ori, oriT, Kan, Cm	5120	Ref. [24]
pJV315	IPTG-inducible GFP, p15A ori, oriT, Kan, Tet	7124	Ref. [24]
pSEVA237R	pBBR1 ori, Kan	3816	SEVA collection [25]
pBBR1-MCS:GFP	GFP production, pBBR1 ori, Kan	5863	Lab stocks
pSEVA351	RSF101 ori, Cm	5120	SEVA collection [25]

## Data Availability

The data presented in this study are available in this article and the Appendix A.

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
