# Peer review of "Genetic Transformation of the Model Quorum Sensing Bacterium Vibrio campbellii by Electroporation"

_genes, 2025, doi:10.3390/genes16060626_

Round 1
Reviewer 1 Report
Comments and Suggestions for Authors
genes-3624857
Title: Genetic Transformation of the Model Quorum Sensing 2 Bacterium Vibrio campbellii by Electroporation
This manuscript presents a detailed and well-executed electroporation protocol for Vibrio campbellii ATCC BAA-1116 and related strains. It addresses a real methodological gap in the genetic manipulation of this crucial marine bacterium, long hampered by reliance on time-consuming conjugation-based transformation techniques. The authors comprehensively optimise parameters, report reproducible transformation efficiencies, and explore factors that may explain variability among strains. Overall, the manuscript is well-written, methodologically rigorous, and engaging to researchers working in microbial genetics, aquaculture, and synthetic biology.
The methods are thoroughly detailed, which ensures that others can reproduce the study. This research offers a strong alternative to conjugation for transforming Vibrio campbellii, a species that is significant both scientifically and in practical applications. The relationship between passage history and the success of transformation and the investigation of extracellular nucleases are particularly intriguing and well-integrated into the discussion.
I have only some minor suggestions to offer:
Abstract: To frame the novelty more strongly, consider briefly stating why electroporation has not previously been widely used in Vibrio campbellii.
Figure clarity: If space allows, a consolidated schematic summarising the optimised protocol (from culture to recovery) could help readers quickly apply the method.
Discussion of Plasmid Host Range: The discussion around plasmid replication origins could be slightly expanded to clarify whether differences in transformation success across strains reflect compatibility rather than efficiency.
Potential Applications: It might strengthen the impact to explicitly mention potential downstream uses of this protocol (e.g. CRISPR, synthetic circuits, aquaculture vaccines).
Typographical edits: A few scattered instances of minor typos or formatting glitches (e.g., spacing before units like “μg DNA”) could be polished during copyediting.
I recommend minor revision, primarily for clarity and small enhancements. This is a solid methodology paper that will likely be well-used by the community.
Reviewer 2 Report
Comments and Suggestions for Authors
Minor Comments
This manuscript presents a newly optimized electroporation protocol for Vibrio campbellii ATCC BAA-1116, achieving transformation efficiencies up to 3×10⁴ CFU/μg DNA. The study systematically evaluates electroporation parameters, cell preparation conditions, and recovery media, identifying optimal settings for successful plasmid uptake. Importantly, the method was partially extended to additional V. campbellii strains and explored the inhibitory role of extracellular nucleases. Proteomic analysis highlighted nuclease expression dynamics potentially linked to transformation resistance. Overall, this work provides a valuable molecular tool for genetic manipulation in V. campbellii and related marine vibrios.
[Lines 12–16]: The authors report a peak transformation efficiency of up to 3 × 10⁴ CFU/μg DNA. What are the contributing factors to variability in electroporation efficiency, and how consistent are these values across independent replicates and plasmid types?
[Lines 18–21]: Transformation success was limited to only three of eight additional V. campbellii strains. Could the authors elaborate on potential genetic or physiological factors—such as restriction-modification systems or membrane composition—that may account for this strain-specific variability?
[Lines 55–58]: Given the salinity-dependent growth requirements of marine vibrios, how does the use of non-ionic osmoprotectants such as sucrose compare to other compatible solutes (e.g., trehalose, betaine) in supporting electroporation efficiency?
[Lines 98–100]: Cells were harvested at OD₆₀₀ ~0.3–0.5. How was this range determined to be optimal, and what are the implications of harvesting at earlier or later growth phases on cell viability and competence?
[Lines 203–205]: The study reports an inverse relationship between plasmid DNA concentration and transformation efficiency. How does this observation deviate from established trends in other Gram-negative bacteria, and what mechanistic hypotheses might explain this inverse correlation?
[Lines 217]: Some strains required up to 180 minutes of recovery post-electroporation to yield transformants. Could the authors discuss the role of cellular recovery kinetics and possible DNA repair mechanisms that may influence transformation outcomes in slow-growing or stressed strains?
[Lines 251–258]: Although 680 mM sucrose is effective for V. natriegens, it impaired transformation in V. campbellii. What physicochemical or cellular factors might explain this discrepancy in tolerance to hyperosmotic buffer conditions during electroporation?
[Lines 259–264]: The type of recovery media showed differential effects across strains. Could the authors provide further biochemical rationale for the observed variation in recovery outcomes using SOC + NaCl versus BHI + v2 salts?
[Lines 289–306]: Despite using standardized parameters, certain V. campbellii and V. harveyi strains failed to yield transformants. To what extent might plasmid-host incompatibility, endogenous nuclease activity, or epigenetic barriers (e.g., methylation patterns) contribute to transformation resistance?
[Lines 309–323]: Proteomic analysis revealed strain-specific trends in nuclease expression over time. What is the significance of DNA-triggered nuclease induction in transformation inhibition, particularly in strain 392 MAV, and how might this response be experimentally validated?
[Lines 327–337]: Homologs of dns were detected in both transformable and non-transformable strains. Given the established role of dns in impeding natural transformation in V. cholerae, do the authors propose targeting this gene via knockout or CRISPRi to enhance transformation in V. campbellii?
[Lines 366–367]: The authors suggest that high DNA concentrations may induce nuclease release. What experimental approaches (e.g., reporter assays, qPCR of nuclease transcripts, or DNase activity assays) could be employed to confirm DNA-induced nuclease expression?
[Lines 393–397]: The potential for linear DNA electroporation and recombineering in V. campbellii is mentioned. What additional genetic tools (e.g., λ Red system, RecET recombinases, counter-selectable markers) would be required to implement efficient homologous recombination in this species?
